# A Complex Hybrid Model for Evaluating Projects to Improve the Sustainability and Health of Regions and Cities

**DOI:** 10.3390/ijerph19138217

**Published:** 2022-07-05

**Authors:** Miroslav Kelemen, Beata Gavurova, Volodymyr Polishchuk

**Affiliations:** 1Faculty of Aeronautics, Technical University of Kosice, 041 21 Košice, Slovakia; miroslav.kelemen@tuke.sk; 2Faculty of Mining, Ecology, Process Control and Geotechnologies, Technical University of Košice, Letná 9, 042 00 Košice, Slovakia; 3Faculty of Information Technologies, Uzhhorod National University, 88000 Uzhhorod, Ukraine; volodymyr.polishchuk@uzhnu.edu.ua

**Keywords:** urban sustainability, healthy cities, projects, risks, expert evaluation, fuzzy sets, European Green Deal, Industry 5.0, decision-making, transport, medical infrastructure

## Abstract

The main goal of the study is to develop a complex hybrid model for evaluating projects to improve the sustainability and health of regions and cities within the European Green Deal and Industry 5.0 concepts. The complex model is a comprehensive evaluation system that considers various influencing factors, the investor’s intentions regarding the need and financing of projects, as well as expert opinion on the possibility of achieving sustainability and health of regions and cities by implementing this project with the investor. The model is based on modern theory of intellectual knowledge analysis, fuzzy set theory, and systems approach. Furthermore, we have an initial quantitative assessment and the linguistic significance of the level of the project financing decision with a reliability assessment. The knowledge from the repository of 896 project plans in the field of transport submitted for implementation and financing in the period 2021–2027 was used for the creation of the model. The results of the study were tested on the examples of evaluation of five real projects and demonstrated the applied value of the methodology for evaluating the level of decision-making feasibility of project financing in uncertainty and the importance of making correct management decisions based on expert opinions.

## 1. Introduction

Society recognizes the importance of addressing the resilience and health of regions and cities in implementing global strategic development plans at the regional and local levels, for example, through the efforts of each EU Member State to reduce regional disparities between the levels of development of European regions and to improve living standards in the least-favored regions [1]. 

The paper, using a multilateral interdisciplinary approach, is an effort to create a mathematical expert hybrid model based on fuzzy logic to support the decision-making processes of evaluation commissions at the state and local government levels, as well as decision-makers for implementing sustainable development policy and healthy regions and cities, improving the governance of cities and regions, in the context of the forthcoming implementation of the European Green Deal and the European Industry 5.0 concept, within the Member States. Addressing the issue also contributes to supporting efforts to become a candidate country and subsequently a full member of the European Union in meeting the EU’s long-term strategic goals and commitments.

The research topic is related to the objective and transparent evaluation of projects based on criteria set by experts in the field to which the call for proposals relates.

The research problem is the quantitative and qualitative evaluation of the proposed solution content and risks in the field of resilience and health of regions and cities within the current societal challenges of climate change elimination expressed in the European Green Deal and the European Industry 5.0 concept.

The research question is focused on researching and creating an innovative algorithm for expert evaluation of projects to strengthen the resilience and health of regions and cities within the European Green Deal and the Industry 5.0 concept submitted in the competition, based on a neuro-fuzzy framework to support decision-making processes. On this basis, the paper proposes Hypotheses 1 and 2.

**Hypothesis** **1** **(H1).***Project developers in strengthening the resilience and health of regions and cities who meet the grant provider’s intentions in accordance with the European Green Deal and Industry 5.0 concept will get a better result based on a fuzzy expert assessment of the project’s innovation ecosystem potential than project developers who do not meet the grant provider’s intentions in the field of strengthening the resilience and health of regions and cities in line with the European Green Deal and the Industry 5.0 concept or only partially meet them*.

**Hypothesis** **2** **(H2).***If the developed algorithm of the comprehensive hybrid model evaluation of resilience and health projects of regions and cities is applicable for the evaluation of projects within the European Union, then the algorithm will be applicable to the evaluation of resilience and health projects of regions and cities outside the European Union*.

The main goal of the study is to develop a complex hybrid model for evaluating projects to improve the sustainability and health of regions and cities, within the concepts of the European Green Deal and Industry 5.0. The model focuses on supporting the decision-making processes of evaluation commissions at the level of public administration and self-government and entities empowered to implement regional policy and local self-government with support for specialized, targeted projects and programs. Secondly, the results of scientific research are also used in the education and support of young scientists and professionals with social responsibility, which is usually expressed in the creation of social responsibility of companies, institutions, or public associations in an open civil society. The common goals are expressed in the intention of the model to maximize the potential of the innovation ecosystem of the project, bringing together different stakeholders in the idea and decision-making process to improve and enhance the sustainability and health of regions and cities according to European Green Deal and Industry 5.0 time, with the accepted level of project implementation risks and the level of required public funds (support from global sources, EU, state support), as well as co-financing of projects from private or corporate sources.

The key result of the paper is a comprehensive approach to evaluating projects to improve the sustainability and health of regions and cities, within the concepts of the European Green Deal and Industry 5.0, for practical use by evaluation commissions and decision-makers at state and local levels private sector. The output is the basis for further research work on the development of a web analytics tool on this topic. The acquired knowledge and algorithm can be transferred to the evaluation of projects outside the European Union, and the problem-solving methodology will allow repeating the procedure to other scientists and experts/evaluators of projects aimed at strengthening the sustainability and health of regions and cities.

### 1.1. Literature Review

The Europe Green Agreement and the Industry 5.0 concept are in line with the global implementation of the 2030 Agenda for Sustainable Development, under the auspices of the United Nations, which reflects the policies and responsibilities of all stakeholders [2].

The EU’s long-term budget, together with the temporary NextGenerationEU instrument (NGEU, €806.9 billion), dedicated to supporting recovery, is the largest stimulus package ever funded in Europe [3].

One third of the €1.8 trillion investment from the NextGenerationEU recovery plan will go to the European Green Agreement and will also be funded by the EU’s seven-year budget to ensure zero net greenhouse gas emissions by 2050, resource-independent economic growth, and no individual or region will be forgotten [4]. Industry 5.0. complements the existing “Industry 4.0” approach by specifically putting research and innovation at the service of the transition to a sustainable, human-centric, and resilient European industry [5].

The implementation of the strategic plans will be supported by various grant schemes for which research and project teams will compete at regional and local level. For example, the budget for financing EU development goals for Slovakia is provisionally at the level of more than 18.6 billion. In addition, Slovakia may have another 7.5 billion at its disposal. EUR for the implementation of the EU Recovery Plan, so the total budget envelope with co-financing of 2.9 billion. The Euro may be at the level of 29.0 billion Euro [6].

Many OECD (Organization for Economic Co-operation and Development) and EU Member States have adopted integrated investment strategies and integrated investment packages as their implementation tool and have put in place mechanisms to coordinate cross-sectoral public investment. More than two thirds of them have developed an integrated national investment strategy like the draft Agenda SK30 (Slovakia’s Development Strategy until 2030) and the National Investment Plan of the Slovak Republic. However, cross-sectoral coordination for investment planning is a major challenge at sub-national level. The lack of cross-sectoral coordination is one of the six main challenges identified by EU Member State authorities, with almost 80% saying it was a major challenge [7]. Based on the analysis presented in the EC’s Eighth Cohesion Report, the main changes in territorial disparities over the last decade and how policies have affected these disparities are set out [8].

Knowledge for solving problems and ideas for projects entered in the competition can also be drawn from the scientific works of the authors, whose results show that climate policy goals can be achieved not only directly but also indirectly by facilitating the implementation of other Sustainable Development Goals (SDGs), these include SDG9 (innovation and infrastructure), SDG5 (gender equality), SDG11 (sustainable cities and communities), and SDG17 (environmental taxation) [9]. Researchers Schwarz et al. [10] recommend building alliances among community, policy, businesses, and science professionals and leveraging connections among organizations, individuals, and agencies that are focused on this agenda, and in adaptive urban planning and design as in Ahern et al. [11]. The findings of scholars, as in the study of Sarwar et al., have significant practical implications for policymakers to introduce economic and non-economic reforms simultaneously to overcome environmental degradation effectively [12]. According to Zang et al., health shocks caused by air pollution are seriously affecting people’s economic lives [13]. 

The work of Saniuk et al. provides an insight into the process of globalization and marginalization of Europe in world production, which prompted the German economy to implement the concept of Industry 4.0 called the Fourth Industrial Revolution [14]. The author Gajdzik examines the need for digitalization of processes in companies moving towards Industry 4.0 [15]. According to Štefko et al., Industry 4.0 and related automation and digitization significantly impact competition between companies. They must deal with the lack of financial resources to apply digital solutions in their businesses [16]. In relation to the Industry 5.0 concept, there is strong scientific interest. For example, the greatest value of selected scientific work of Madddikunta et al. [17] is that it discusses the perspectives of smart healthcare, cloud manufacturing, supply chain management, manufacturing, and various other applications as application development to be developed and run in Industry 5.0. Xu et al. [18] emphasize the importance and values of the transition from Industry 4.0 to Industry 5.0. The aim of another scientific study of Sindhwani at al. [19] is to analyze the factors of Industry 5.0 to achieve sustainability by integrating human values with technology. Moreover, the following work of Yin et al. [20] examines and proves the need to create digital green knowledge in the Industry 5.0 concept.

The empirical results of the studies can contribute to supporting environmental and economic policies in EU countries in achieving their sustainable development and the objectives of the European Green Agreement, as well as the sustainability of small and medium-sized enterprises [21,22,23,24]. Ferre-Comala et al. [25] represent the first step in introducing fuzzy logic into models of economic growth, respectively. In the work of Valla-Lloser et al. [26] we find data analyzed using multimethod models estimated as models of structural equations with mean and covariance structure. Vochozka et al. examine IoT-based systems of things, sustainable Industry 4.0 wireless networks, and digitized mass production in cyber-physical intelligent production [27]. As innovation enables Small and Medium-Sized Enterprises (SMEs) to be more competitive with their competitors, more innovative activities can force SMEs to overcome these challenges [28]. According to Tobisova et al. investment is a challenging and threatening indicator for businesses not only in times of depression, such as the current coronavirus pandemic, but also under normal market conditions. They present the methodology of financial risk assessment and investment development as an effective tool for corporate sustainability [29]. Optimization of financial costs to maintain the reliability of systems, assuming the estimated level of optimization of unexpected costs for the operator, is a crucial parameter [30]. Rehak et al. draw attention to the critical infrastructure system, which identifies the subsystems necessary for the functioning of the state (such as energy, transport, and emergency services). These issues are important for the resilience and health of the state and the regions [31]. In the work of Haškova et al. (2021), we find inspiring ideas for using the advantages of fuzzy logic in project evaluation, when we often encounter uncertainty in connection with randomness or ambiguity [32]. These tools have also been used innovatively in the field of safe and sustainable air transport [33] or in the expert evaluation of selected components of the smart city concept [34].

However, to date, no comprehensive approach to evaluating projects to improve the resilience and health of regions and cities based on a fuzzy set theory using hybrid methods has been implemented. In response to all these facts, it was decided to conduct a topical study to develop a complex hybrid model for evaluating projects to improve the sustainability and health of regions and cities, within the concepts of the European Green Deal and Industry 5.0. The hybrid integrated model determines the level of funding opportunities for the project, considering the target needs of investors and expert opinions on the possibility of achieving goals to improve the sustainability and health of regions and cities through the implementation of this project. The hybrid integrated model focuses on the unbiased evaluation of grant applicants and increases the security of their funding.

The comprehensive model is a complex system of functioning that considers various factors of influence, such as the importance of the project idea to improve the sustainability and health of regions and cities; risk-oriented factors of influence that potentially lead to the success of the project; factors of human influence and the team of project implementers, their experience, and knowledge in the field of sustainability and health of regions and cities. It also considers the investor’s goals regarding the need and possibility of financing projects, as well as expert opinion on the possibility of achieving the goal of improving the sustainability and health of regions and cities by implementing this project with the support of investors. In addition, to adequately support the decision-making and processing of information obtained from experts, the model is based on the modern theory of intellectual analysis of knowledge, fuzzy set theory, and a systems approach. 

### 1.2. Formal Formulation of the Evaluation Problem

Today, many innovative projects address the challenges of increasing the sustainability and health of regions and cities. The implementation of such projects allows to reduce the negative impact of human activities effectively and quickly on the environment, improve monitoring and improve the health of citizens. It is no secret that start-up (innovative) projects or grant projects provide faster and better solutions than government projects that involve complex bureaucratic actions and procedures. In addition, there are many problems in an emergency, the solution of which is important and necessary in a critically short period of time. Proof of this is the work of the municipality/region/state in the context of the COVID-19 pandemic. For example, the amount of garbage in quarantine has increased by an average of six times, as well as the big problem of disposing of personal protective equipment and others.

The resilience of regions and cities is characterized by biodiversity, versatility, multilevel networks, modularity, and adaptive design [11], as well as not only the concept of prevention and mitigation of regional and urban disasters (crisis situations) [35]. In this context, the following definitions are given for our study:

**Definition** **1.***Projects to increase the sustainability and health of regions and cities are projects based on innovative technologies and aimed at reducing the negative impact of human activities on the environment and/or improving the health of citizens, within the concepts of the European Green Deal and Industry 5.0. These are projects that affect sociocultural, political, economic, environmental, security and health, decision-making processes, policies, and activities aimed at improving health, lifestyle, equality, and equity of affordable health, services, and tools in regions and cities*. 

**Definition** **2.***Sustainability of regions (cities) is the ability of regions (cities) to withstand unknown risks and recover from a disaster*.

The resilience of regions and cities is also inherently perceived in connection with public health issues. Sometimes these projects are also presented as so-called SMART Region or SMART City projects, investment project packages, etc. Objective and transparent expert evaluation of projects, especially in the specific area of strengthening the resilience and health of regions and cities within the European Green Deal and the Industry 5.0 concept, requires experience and complexity in evaluation. There is a need to finance such projects for their implementation and introduction on the market. Financing innovative projects is a risky activity. To minimize risks, it is necessary to have adequate decision support systems for evaluating the projects themselves, the teams implementing the project, and the risks. Therefore, the model is called hybrid, because, on the one hand, it uses project data from the project application, which is structured, poorly structured, or unstructured, and on the other hand combines the experience and knowledge of experts. The obtained level, which is a complex indicator, increases the degree and guarantees the security of financing of such projects.

Since the task for the solution is the area of expert evaluation, the following subjects of management are presented: Experts—persons who analyze and evaluate the project application; Investors are entities that are willing to finance evaluated projects; Project analyst is a person who adjusts the whole evaluation process, considering the needs of investors. 

Let the system theoretical-multiple model problem of project evaluation to improve the sustainability and health of regions and cities, within the concepts of the European Green Deal and Industry 5.0, be presented as follows:(1)$$\left\{P,K_{P},K_{R},K_{T},M_{P},M_{R},M_{T},G_{P},G_{R},G_{T},L,M_{A}|Y\left(f\right)\right\},$$
where:
$P=\left(p_{1};p_{2};\ldots ;p_{n}\right)$—a set of projects submitted to some experts for funding by investors;$K_{P}$—information models of criteria (groups of criteria) to assess the importance of the project idea to improve the sustainability and health of regions and cities;$K_{R}$—information models of criteria for assessing risk-oriented factors of influence that will potentially lead to the failure of the project;$K_{T}$—information models of criteria (groups of criteria) for assessing human factors and the team of project implementers, their experience, and knowledge in the field of sustainability and health of regions and cities;$M_{P}$—fuzzy project evaluation model to improve the sustainability and health of regions and cities; $M_{R}$—fuzzy model for assessing the risks of project implementation to improve the sustainability and health of regions and cities; $M_{T}$—fuzzy model for assessing the competencies of the project implementation team;$G_{P}$—the goal of the importance of the region where the project will be implemented; $G_{R}$—the goal of acceptable risks;$G_{T}$—the goal of the competence of the project implementation subjects; L—expert opinions on the possibility of achieving the goal of improving the sustainability and health of regions and cities, by implementing this project with the support of investors and considering their goals;$M_{A}$—the model of aggregation of output data for deriving the level of decision-making expediency of project financing.

As a result, the output estimate is obtained $f=\mu_{Y}\left(f\left(\varphi_{e}\right)\right)\;$ and level Y, which contains the feasibility of project financing, within the concepts of the European Green Deal and Industry 5.0, considering the goal needs of investors G  and expert opinions L, thereby increasing funding projects security.

The complex hybrid model of project evaluation is shown in the form of a block diagram (Figure 1).

This study is arranged as follows: In Section 2, we describe the formal formulation of the problem, the hybrid complex model, and the information models for input processing. In Section 3, we will outline a simulation experiment, perform verification, and test the developed model on the example of five project evaluation. In Section 4, we discuss the results of the study and the advantages and disadvantages of the developed hybrid model. In Section 5, we conclude and present the main results that have been achieved for the first time. Ideas for future work and improvements are presented in conclusions of paper, namely the development of information technology and its support in the form of software for project evaluation with the specific focus.

## 2. Materials and Methods

The stages of designing a complex hybrid model will be presented in terms of the fuzzy mathematical models for obtaining project estimates for different information models, and the model of aggregation of initial estimates to determine the level of project funding. 



$M_{P}$

*—fuzzy project evaluation model to increase the sustainability and health of regions and cities.*


This model is based on the opinions of experts in a range of values of project quality for certain indicators. At the first stage of X evaluation projects, the expert analyzes the project application based on his own experience and expresses his views according to $K_{P}$—information model of criteria (groups of criteria) to assess the importance of the project idea to improve the sustainability and health of regions and cities. Information indicators $K_{P}=\left(K_{11},K_{12},\ldots ,K_{ij},\ldots ,K_{ck_{c}}\right)$ are divided into several groups $C=\left(C_{1},C_{2},\ldots ,C_{c}\right)$. Without reducing the generality and visual presentation of the material, we will consider one project for this model. The calculations are similar for any number of projects. 

In practice, experts most often express their opinions not in clear discrete values, but in intervals, for example: “project evaluation according to a certain criterion should be close to a certain number”, or “project evaluation according to a certain criterion is likely, to be within certain limits”, etc. Therefore, it is advisable to estimate *x* for each criterion in some range of numbers $x\in \left[a_{0};b_{0}\right]$. The value of the interval is set by the project analyst of the organization that evaluates projects.

For the study, 7 experts with more than 15 years of experience in project management were invited to express their opinion on some criterion $K_{c}$ according to one of the fuzzy statements *RL* = {*A*; *B*; *C*; *D*}, where: *A* = {project evaluation according to a certain criterion is in the range from $a_{1}$ to $b_{1}$}; *B* = {project evaluation according to a certain criterion is close to the number $b_{1}$}; *C* = {project evaluation according to a certain criterion is not greater than the number $b_{1}$}; *D* = {project evaluation according to a certain criterion is not less than the number $b_{1}$}. Set fuzzy statements are open. If necessary, the project analyst can supplement or change it.

Next, all fuzzy statements will be described using the model of presentation of fuzzy knowledge [36], using the functions of belonging criteria.

Fuzzy statement *A =* {project evaluation according to a certain criterion is in the range from $a_{1}$ to $b_{1}$} consider it as a fuzzy set and describe it using the trapezoidal membership function:(2)$$\mu_{A}\left(x\right)=\left\{\begin{array}{cc} 0, & x\leq a_{0};\\ \frac{x-a_{0}}{a_{1}-a_{0}}, & a_{0}\leq x<a_{1};\\ 1, & a_{1}\leq x<b_{1};\\ \frac{b_{0}-x}{b_{0}-b_{1}}, & b_{1}\leq x<b_{0};\\ 0, & x\geq b_{0}. \end{array}\right.$$

Statement *B* = {project evaluation according to a certain criterion is close to the number $b_{1}$}, correctly describe using the triangular membership function: (3)$$\mu_{B}\left(x\right)=\left\{\begin{array}{cc} 0, & x\leq a_{0};\\ \frac{x-a_{0}}{b_{1}-a_{0}}, & a_{0}<x\leq b_{1};\\ \frac{b_{0}-x}{b_{0}-b_{1}}, & b_{1}<x<b_{0};\\ 0, & x\geq b_{0}. \end{array}\right.$$

Next, consider the statement *C* = {project evaluation according to a certain criterion is not greater than the number $b_{1}$}, which is characterized in fuzzy sets by the *Z*-shaped type of membership function. Here we offer a *Z*-line:(4)$$\mu_{C}\left(x\right)=\left\{\begin{array}{cc} 1, & 0<x\leq b_{1};\\ \frac{b_{0}-x}{b_{0}-b_{1}}, & b_{1}<x\leq b_{0};\\ 0, & x>b_{0}. \end{array}\right.$$

Statement *D* = {project evaluation according to a certain criterion is not less than the number $b_{1}$} represents an uncertainty of type *S*-like membership function, we propose an *S*-linear function:(5)$$\mu_{D}\left(x\right)=\left\{\begin{array}{cc} 0, & x\leq a_{0};\\ \frac{x-a_{0}}{b_{1}-a_{0}}, & a_{0}<x\leq b_{1};\\ 1, & x>b_{1}. \end{array}\right.$$

Thus, the statements of experts on project indicators will not be discrete values, but intervals of values, according to the chosen statement *RL* = {*A*; *B*; *C*; *D*}.

At the next stage, we will pass from intervals of indistinct estimations of the project according to criteria of one group, to one-point estimation within group $C=\left(C_{1},C_{2},\ldots ,C_{c}\right)$. 

To do this, let the design analyst set the weights for each evaluation criterion $w_{ij},\;i=\bar{1,c};j=\bar{1,k_{c}}$, for example, from the interval $\left[1;10\right]$. If there is no need to set weights, then we can take them as equally important. For further calculations, we perform their rationing within the relevant group:(6)$$\alpha_{ij}=\frac{w_{ij}}{\sum_{j=1}^{k_{c}}w_{ij}},\;i=\bar{1,c};\;\;\alpha_{ij}\in \left[0;1\right],$$
where the condition is met $\sum_{j=1}^{k_{c}}\alpha_{ij}=1$.

To fuzzification the data, we will find a weighted sum to determine the aggregate expert opinion within some group of criteria $C=\left(C_{1},C_{2},\ldots ,C_{c}\right)$:(7)$$\bar{\varepsilon_{i}}\left(x\right)=\overset{k_{c}}{\underset{j=1}{\sum }}\mu_{RL\;ij}\left(x\right)\cdot \alpha_{ij},\;i=\bar{1,c}.$$

Thus, they were obtained for each group $C_{1},C_{2},\ldots ,C_{c}$ criteria a continuous set of values $\bar{\varepsilon_{1}}\left(x\right),\;\bar{\varepsilon_{2}}\left(x\right),\ldots ,\bar{\varepsilon_{c}}\left(x\right);\;\bar{\varepsilon }\left(x\right)\in \left[0;1\right]\;$ on the interval of evaluation points $x\in \left[a_{0};b_{0}\right].$ For the aggregate assessment of the expert opinion $x_{i}\in \left[a_{0};b_{0}\right]$ of the corresponding group of criteria $i=\bar{1,c}$, the maximum value of the membership function of the weighted sum has been collected $\underset{x}{\max}\bar{\varepsilon_{i}}\left(x\right)$. Thus, we obtain an aggregated conclusion of the expert’s opinions on the criteria for each group $(x_{1},x_{2},\ldots ,x_{c})\in \left[a_{0};b_{0}\right]$. That is, for a group of criteria that have a common meaning, from the expert opinion, expressed in the range of values, the aggregate conclusion was reached, which is a numerical value from the range $\left[a_{0};b_{0}\right]$.

In the third stage, the aggregated conclusions of the experts on the groups of criteria will be combined into a general assessment, which will represent the normalized value of the level of the project idea to improve the sustainability and health of regions and cities. To do this, let the project analyst set the weights for each group of criteria $C=\left(C_{1},C_{2},\ldots ,C_{c}\right)$, $w_{i},\;i=\bar{1,c}$, for example, from the interval $\left[1;10\right]$. If there is no need to set weights for groups of criteria, then we can take them as equally important. For further calculations we carry out their rationing:(8)$$\alpha_{i}=\frac{w_{i}}{\sum_{i=1}^{c}w_{i}},\;i=\bar{1,c};\alpha_{i}\in \left[0;1\right].\;$$

For data defuzzification, we construct an aggregate estimate using the convolution model. For example, take a weighted average convolution:(9)$$m_{p}=\frac{1}{b_{0}}\overset{c}{\underset{i=1}{\sum }}x_{i}\cdot \alpha_{i}.\;$$

The obtained estimate $m_{p}\in \left[0;1\right]$ characterizes the importance of the level of the project idea to improve the sustainability and health of regions and cities. The higher the score, the better the project application and good opportunities for its implementation. 

Estimates of other projects from the set are calculated similarly $P=\left(p_{1};p_{2};\ldots ;p_{n}\right)$.

Thus, from the linguistically vague conclusions of the expert to the input descriptive (textual) data obtained from the project application to the quantitative assessment was transferred. Adequate use of the apparatus of fuzzy sets increases the degree of validity of future decisions. 

$M_{R}$—*fuzzy model for assessing the risks of project implementation to improve the sustainability and health of regions and cities*.

The authors have already developed several fuzzy project risk assessment models for both investment and innovation projects, but they have not been verified for the evaluation of sustainability and health projects in regions and cities.

Let the set of projects $P=\left(p_{1};p_{2};\ldots ;p_{n}\right)$, be submitted to some experts for funding by their investors. Projects will be evaluated according to the proposed information model of criteria for assessing risk-oriented factors influencing $K_{R}$, according to the criteria $\tilde{K_{R}}=\left(K_{R1},K_{R2},\ldots ,K_{Rk_{R}}\right)$.

Each risk-oriented impact factor (risk criterion) will be assessed in a hybrid way, namely:Conclusions on the level of probability of occurrence of the risk situation described by the relevant criterion $\tilde{K_{R}}$. We propose to unify such conclusions with the help of one of the terms of the following term set: *T* = {$t_{1}$(low level of risk); $t_{2}$(below average level of risk); $t_{3}$(average level of risk); $t_{4}$(above average level of risk); $t_{5}$(high level of risk)};Number confidence Δ of the expert’s reasoning from the interval [0; 1], for each conclusion according to the corresponding criterion $\tilde{K_{R}}$. Assuming the following content: 0—minimum confidence in their conclusions, and 1, respectively—maximum.

We present a fuzzy model for assessing the risks of project implementation in the form of an operator:(10)$$\epsilon \left(t;\Delta \right)\to m_{r},$$
where ϵ—the operator that corresponds to the initial normalized value of the risks of the project $m_{r}$, with input variables t;Δ. 

Here are the stages to obtain the initial estimate $m_{r}$ for some project *p*. In the case of multiple projects, the evaluation procedure will be repeated for all projects. 

At the first stage we will carry out fuzzification of input hybrid data. 

Let the term set of linguistic variables *T* be represented on some numerical interval, for delimitation of terms $\left[a_{1};a_{6}\right]$, where $t_{1}\in \left[1;20\right],$ $T_{1}\in \left[a_{1};a_{2}\right],$ $t_{2}\in \left(20;40\right],$ $T_{2}\in \left[a_{2};a_{3}\right],$ $T_{3}\in \left[a_{3};a_{4}\right],$ $T_{3}\in \left[a_{3};a_{4}\right]$. The presented numerical interval is argued by the fact that the indicators characterize the level of probability of a risk event. Then it is natural to consider this level of risk as a percentage. If necessary, the interval partition values can be adjusted and changed by the project analyst. 

To fuzzification the input hybrid data $\left(t_{u};\Delta_{u}\right),\;u=\bar{1,k_{R}}$ use the model of presentation of fuzzy knowledge in multidimensional space [4]. We use the cone-like membership function of belonging of two variables to combine conclusions on the level of probability of occurrence of a risk situation and the number confidence of the expert’s opinions on providing his opinion:(11)$$\mu \left(K_{Ru}\right)=\left\{\begin{array}{c} 1-\varepsilon_{u},\;if\;\varepsilon_{u}<1,\\ \;0,\;\;\varepsilon_{u}\geq 1. \end{array}\right.\;O_{u}=\left\{\begin{array}{cc} 100-20\;\cdot \;\Delta_{u}, & \mathrm{if}\;\mathrm{the}\;\mathrm{conclusion}\;t_{1};\\ 100-40\;\cdot \;\Delta_{u}, & \mathrm{if}\;\mathrm{the}\;\mathrm{conclusion}\;t_{2};\\ 100-60\;\cdot \;\Delta_{u}, & \mathrm{if}\;\mathrm{the}\;\mathrm{conclusion}\;t_{3};\\ 100-80\;\cdot \;\Delta_{u}, & \mathrm{if}\;\mathrm{the}\;\mathrm{conclusion}\;t_{4}.\\ 100-100\;\cdot \;\Delta_{u}, & \mathrm{if}\;\mathrm{the}\;\mathrm{conclusion}\;t_{5}. \end{array}\right.\;O_{i}=\left\{\begin{array}{ccc} a_{2}\cdot q_{i}, & \mathrm{if} & t_{i}\in T_{1};\\ a_{3}\cdot q_{i}, & \mathrm{if} & t_{i}\in T_{2};\\ a_{4}\cdot q_{i}, & \mathrm{if} & t_{i}\in T_{3};\\ a_{5}\cdot q_{i}, & \mathrm{if} & t_{i}\in T_{4};\\ a_{6}\cdot q_{i}, & \mathrm{if} & t_{i}\in T_{5}. \end{array}\right.\;\varepsilon_{u}=\sqrt{\frac{{\left(O_{u}-100\right)}^{2}}{100^{2}}+{\left(\Delta_{u}-1\right)}^{2}}$$

$O_{u},\;\Delta_{u}$—the value of the *u*-th criterion $u=\bar{1,k_{R}}$.

The content of the membership function $\mu \left(K_{Ru}\right)$ shows the level of the criterion, is the greater the value of $\mu \left(K_{Ru}\right)$ the higher the level. 

Without reducing the generality, the project analyst can also use other known methods to aggregate qualitative data. Presentation of input data in the form of linguistic assessment and the certainty of its assignment allows you to better reveal the views of experts. 

In the second stage, the project analyst introduces weighting factors for each risk-oriented factor of influence (criterion) on the project implementation. 

Denote the weights $v_{u}$, $u=\bar{1,k_{R}}$, from some interval [1; 10]. Otherwise, risk criteria may be equally important. Because it works in the space of assessments [0; 1], then, similarly, you need to normalize the weights:(12)$$\bar{v_{u}}=\frac{v_{u}}{\sum_{u=1}^{k_{R}}v_{u}},u=\bar{1,k_{R}}.$$

In the third stage, we derive an aggregate risk assessment. To do this, build a membership function, as one of the proposed convolutions, depending on the wishes of the project analyst. Without reducing the generality, for example, take the middle convolution:(13)$$m_{r}=\sum_{u=1}^{k_{R}}\bar{v_{u}}\cdot \mu \left(K_{Ru}\right).$$

The obtained value has the following meaning: the larger the aggregate estimate $m_{r}\in \left[0;1\right]$, the lower the risks of the project. 

Thus, from the conclusions on the level of probability of the risk situation and the number of confidences of the expert’s opinions on this issue, a quantitative assessment has been made, which increases the degree of validity of future decisions. Estimates of other projects from the set are calculated similarly $P=\left(p_{1};p_{2};\ldots ;p_{n}\right)$.

The advantages of the model are argued by the fact that based on input; hybrid data reveals the vagueness of input estimates. Improves the efficiency of obtaining input assessments using the experience, knowledge, and expertise of experts. The model can derive a quantitative normalized output value of the risks of the project, which increases the degree of validity of further management decisions. 

$M_{T}$*—fuzzy model for assessing the competencies of the project implementation team*.

Here it is proposed to use one of the models already developed by the authors, which at the output we obtain the normalized value $m_{t}\in \left[0;1\right]$, o separately for the set of projects *P*, namely: Information model of evaluation and output rating of start-up projects development teams; Model of evaluation and selection of the expert group members; Model of evaluation and selection of expert group members for Smart Cities, Green Transportation and Mobility: from safe times to pandemic times [34].

Thus, normalized estimates of projects were obtained $m_{p}\left(x_{e}\right),\;m_{r}\left(x_{e}\right),\;m_{t}\left(x_{e}\right),\;e=\bar{1,n}$, according to the proposed fuzzy evaluation models $M_{P},\;M_{R},\;M_{T}$.

The peculiarity of the complex hybrid model is that the investor has some goals in terms of the need and ability to finance projects to improve the sustainability and health of regions and cities. Such goals are correlated according to the models $M_{P},\;M_{R},\;M_{T}$. This is because in the future we will look for the closest distance between the estimates of models and objectives. Therefore, they must have the same meaning. Based on the experience of the authors in the subject area, the following goals are proposed:

$G_{P}$—the goal of the importance of the region where the project will be implemented.

Here we propose to use the categorization of regions, according to the Ministry of Transport and Construction of the Slovak Republic, as follows:Category 1—regions of international importance;Category 2—regions of national importance (national);Category 3—regions of supraregional importance;Category 4—regions of predominant importance at the regional level.

Without reducing the generality, the categories of project regions analysts can easily change according to needs and conditions. 

$G_{R}$—the goal of acceptable risks is the level of risk that an investor can afford by investing in a project to increase the sustainability and health of regions and cities.

$G_{T}$—the goal of the competence of the project implementation subjects.

Since the problem of evaluating alternative projects consists of three goals, then the vectors $m_{p}\left(p\right)=\left\{m_{p}\left(p_{1}\right),\;m_{p}\left(p_{2}\right),\ldots ,m_{p}\left(p_{n}\right)\right\}$, $m_{r}\left(p\right)=\left\{m_{r}\left(p_{1}\right),\;m_{r}\left(p_{2}\right),\ldots ,m_{r}\left(p\right)\right\}$ and $m_{t}\left(p\right)=\left\{m_{t}\left(p_{1}\right),\;m_{t}\left(p_{2}\right),\ldots ,m_{t}\left(p_{n}\right)\right\}$, we project on the three-dimensional coordinate system x, y, z respectively. For each project we get the coordinates of the goals $G_{P},G_{R}$,$\;G_{T},$ which we present in the form: $\left(m_{p}\left(p_{1}\right),\;m_{r}\left(p_{1}\right),\;m_{t}\left(p_{1}\right)\right)$, $\left(m_{p}\left(p_{2}\right),\;m_{r}\left(p_{2}\right),\;m_{t}\left(p_{2}\right)\right),$…, $\left(m_{p}\left(p_{n}\right),\;m_{r}\left(p_{n}\right),\;m_{t}\left(p_{n}\right)\right)$.

Furthermore, a three-dimensional vector of investor goals was considered $T^{*}=\left(A_{1},A_{2},A_{3}\right),$ which considers the wishes of investors regarding the importance of alternative projects according to the goals $G_{P},G_{R}$,$\;G_{T}$. The vector of investor goals was modeled as follows [37]. 

Let us analyze an object with 3 inputs and one output:(14)$$U=\left(A_{1},A_{2},A_{3}\right),$$
where U—the vector of the initial estimate $\left(u_{1},u_{2},u_{3}\right)$, and its components may have values from the interval [0; 1], and $A_{1},A_{2},A_{3}$ are input linguistic variables.

To evaluate the linguistic variables $A_{1},A_{2},\;A_{3}$ qualitative terms from such term sets were used:(15)$$A_{1}=\left(a_{11},a_{12},\ldots ,a_{1t}\right),A_{2}=\left(a_{21},a_{22},\ldots ,a_{2t}\right),\;A_{3}=\left(a_{31},a_{32},\ldots ,a_{3t}\right).$$

Knowledge of the vector of investors’ goals $T=\left(t_{1},t_{2},t_{3}\right)$ is obtained from the base of fuzzy knowledge, consisting of systems of logical expressions—“If—Then, Else”, which link the values of input variables $A_{1},A_{2},A_{3}$ with one of the possible values U.
(16)$$\mathrm{IF}\;A_{1}=a_{1t}\;\mathrm{and}\;A_{2}=a_{2t}\;\mathrm{and}\;A_{3}=a_{3t}\;\mathrm{THEN}\;U=\left(u_{1},u_{2},u_{3}\right)\;\mathrm{ELSE}\ldots$$

Thus, the project analyst sets the linguistic wish of the vector of investors’ goals, which is translated into the vector of the initial quantitative and normalized assessment $\left(u_{1},u_{2},u_{3}\right)$, which is denoted accordingly $\left(u_{1},u_{2},u_{3}\right)=\left(t_{1},t_{2},t_{3}\right)$.

The vague knowledge base for project evaluation to improve the sustainability and health of regions and cities is offered as follows:

**IF** we have goals:
$G_{P}$—the goal of the importance of the region where the project will be implemented:$a_{11}$ there are 4 categories of the region then $u_{1}=0.4$;$a_{12}$ there are 3 categories of the region then $u_{1}=0.6$;$a_{13}$ there are 2 categories of the region then $u_{1}=0.8$;$a_{14}$ there are 1 category of the region then $u_{1}=1$.

**AND** $G_{R}$—the goal of acceptable risks:
$a_{21}$ high risk then $u_{2}=0.2$; $a_{22}$ average risk then $u_{2}=0.4$;$a_{23}$ low risk then $u_{2}=0.6$;$a_{24}$ very low risk then $u_{2}=0.8$;$a_{25}$ minimal risk then $u_{2}=1.$

**AND** $G_{T}$—the goal of the competence of the project implementation subjects:
$a_{31}$ not interested in competence then $u_{3}=0.2$; $a_{32}$ may even be low competencies then $u_{3}=0.5$;$a_{33}$ are interested in average competencies then $u_{3}=0.7$;$a_{34}$ need the best competencies then $u_{3}=1$.

**THEN** logical statement can be formulated as follows: 

If the investor needs the importance of the region where the project will be implemented $A_{1}$, the acceptable risk $A_{2}$ and the competence of the project implementation subjects $A_{3}$ then $U=\left(u_{1},u_{2},u_{3}\right)$.

A project analyst can change quantitative levels, or rules in goals. Therefore, the knowledge base is open, and the number of goals can be increased if necessary.

For all projects we find the values $Z_{e}=\left(z_{pe},z_{re},z_{te}\right),e=\bar{1,n}$, which characterize the relative estimates of the proximity of the evaluated projects to the vector of investor goals for each goal $G_{P},G_{R},G_{T}$, removing the question of different rating scales [37]:(17)$$z_{pe}=1-\frac{\left|u_{1}-m_{p}\left(p_{e}\right)\right|}{max\left\{u_{1}-\underset{e}{min}\;m_{p}\left(p_{e}\right);\;\underset{e}{max\;}m_{p}\left(p_{e}\right)-u_{1}\right\}},$$
(18)$$z_{re}=1-\frac{\left|u_{2}-m_{r}\left(p_{e}\right)\right|}{max\left\{u_{2}-\underset{e}{min}\;m_{r}\left(p_{e}\right);\underset{e}{\;max}\;m_{r}\left(p_{e}\right)-u_{2}\right\}},$$
(19)$$z_{te}=1-\frac{\left|u_{3}-m_{t}\left(p_{e}\right)\right|}{max\left\{u_{3}-\underset{e}{min}\;m_{t}\left(p_{e}\right);\;\underset{e}{max}\;m_{t}\left(p_{e}\right)-u_{3}\right\}}.$$

If one project is submitted for evaluation, then the investor does not need to comment on their own goals and this stage is skipped. To find the vector of values $Z_{e}$ for projects must be at least two projects.

Furthermore, to aggregate the values of $Z_{e}$, it is proposed to use modeling of uncertainties of the form “average value” in three-dimensional space, using the cone-like membership function in the estimation space [0; 1]. Moreover, the value of the center of the base of the cone will be a unit vector $\left(x_{1}^{0};x_{2}^{0};x_{3}^{0}\right)=\left(1;1;1\right)$, and the experimentally obtained scaling by the coordinates of the vector $Z_{e}$ will be $\left(3;3;3\right)$. Then, the three-dimensional cone-like membership function will be given by the formula: (20)$$\varphi_{e}=\left\{\begin{array}{c} 1-\vartheta_{e},\;\;if\;\vartheta_{e}<1,\\ 0,otherwise. \end{array},\right.\;\mathrm{where}:\;\vartheta_{e}=\frac{1}{3}\cdot \sqrt{{\left(z_{pe}-1\right)}^{2}+{\left(z_{re}-1\right)}^{2}+{\left(z_{te}-1\right)}^{2}},e=\bar{1,n}.\;$$

Thus, the initial estimates $\varphi_{e}$ will be obtained from the interval [0; 1] on n projects to increase the sustainability and health of regions and cities. The vector of investors’ goals provides the construction of a ranking of alternatives given by the vectors of assessments and increases the security of choosing alternatives according to the target needs. The initial assessment is based on the assessment of the importance of the project idea, potential risks of the project, the competence of development teams, and considers the goals of investors on the importance of the region where the project will be implemented, risk acceptability, competence of the project implementation subjects. 

Next, let the experts evaluating the projects express their conclusions on the possibility of achieving the goal of improving the sustainability and health of regions and cities by implementing this project with the support of investors and considering their goals. For this conclusion we introduce the linguistic variable $L=\left\{L_{1};\;L_{2};\ldots ;L_{5}\right\}$, where: $L_{1}$—high possibility of project implementation taking into account the goals of investors; $L_{2}$—the possibility of project implementation taking into account the goals of investors above average; $L_{3}$—average possibility of project implementation taking into account the goals of investors; $L_{4}$—low possibility of project implementation taking into account the goals of investors; $L_{5}$—very low possibility of project implementation taking into account the goals of investors. 

Next, $M_{A}$—the model of aggregation of output data is proposed for deriving the level of decision-making expediency of project financing. A graphical interpretation of the $M_{A}$ model is presented in Figure 2.

To interpret the dependence of the initial assessment $\varphi_{e}$ and expert opinion L on the possibility of achieving the goal of improving the sustainability and health of regions and cities, by implementing this project with the support of investors, we offer the following membership function:(21)$$f\left(\varphi_{e}\right)=\left\{\begin{array}{cc} 0, & \varphi_{e}<0;\\ {\left(\varphi_{e}\right)}^{k}, & 0\leq \varphi_{e}<1;\\ 1, & \varphi_{e}\geq 1. \end{array}\right.\;e=\bar{1,n},$$
where *k* is the threshold for the possibility of achieving the project goal with the support of investors’ goals. The value of this threshold varies depending on the expert opinion L. This threshold can be obtained by learning from the test data of projects, having a history of projects, and investigating errors of the first and second kind. Mistakes of the first kind occur in cases where the project has good performance, receives funding, and the project does not implement. The second kind of error occurs when the project does not receive funding but implements it in another way. For example, we experimentally set: *k* = $\frac{2}{9}$ when we have an expert opinion $L_{1}$; *k* = $\frac{7}{9}$ when we have an expert opinion $L_{2}$; *k* = $\frac{4}{9}$—expert opinion $L_{3}$; *k* = $\frac{5}{9}$—expert opinion $L_{4}$; *k* = $\frac{3}{2}$—expert opinion $L_{5}$. 

Thus, we obtained aggregated normalized estimates $f\left(\varphi_{e}\right),\;e=\bar{1,n}$ from the interval [0; 1], on project evaluation models, investor goals and expert opinions that evaluate projects.

Levels Y of feasibility of project financing taking into account goals of investors and conclusions of experts, we will present as follows: $y_{1}$—very low level of feasibility of project financing; $y_{2}$—low level of feasibility of project financing; $y_{3}$—average level of feasibility of project financing; $\;y_{4}$—high level of feasibility of project financing; $y_{5}$—very high level of feasibility of project financing.

Levels of decision-making Y are correctly considered using triangular membership functions. This is because they will have intersections of the original values, and this will expand the ability to make decisions:(22)$$\mu_{y_{1}}\left(f\left(\varphi_{e}\right)\right)=\left\{\begin{array}{cc} 1, & f\left(\varphi_{e}\right)\leq \delta -\frac{\delta }{2};\\ \frac{3\delta -4\;\cdot \;f\left(\varphi_{e}\right)}{\delta }, & \delta -\frac{\delta }{2}<f\left(\varphi_{e}\right)\leq \delta -\frac{\delta }{4}. \end{array}\right.$$
(23)$$\mu_{y_{2}}\left(f\left(\varphi_{e}\right)\right)=\left\{\begin{array}{cc} \frac{4\;\cdot \;f\left(\varphi_{e}\right)-2\delta }{\delta }, & \delta -\frac{\delta }{2}<f\left(\varphi_{e}\right)\leq \delta -\frac{\delta }{4};\\ \frac{4\delta -4\;\cdot \;f\left(\varphi_{e}\right)}{\delta }, & \delta -\frac{\delta }{4}<f\left(\varphi_{e}\right)\leq \delta . \end{array}\right.$$
(24)$$\mu_{y_{3}}\left(f\left(\varphi_{e}\right)\right)=\left\{\begin{array}{cc} \frac{4\;\cdot \;f\left(\varphi_{e}\right)-3\delta }{\delta }, & \delta -\frac{\delta }{4}<f\left(\varphi_{e}\right)\leq \delta ;\\ \frac{5\delta -4\;\cdot \;f\left(\varphi_{e}\right)}{\delta }, & \delta <f\left(\varphi_{e}\right)\leq \delta +\frac{\delta }{4}. \end{array}\right.$$
(25)$$\mu_{y_{4}}\left(f\left(\varphi_{e}\right)\right)=\left\{\begin{array}{cc} \frac{4\;\cdot \;f\left(\varphi_{e}\right)-4\delta }{\delta }, & \delta <f\left(\varphi_{e}\right)\leq \delta +\frac{\delta }{4};\\ \frac{6\delta -4\;\cdot \;f\left(\varphi_{e}\right)}{\delta }, & \delta +\frac{\delta }{4}<f\left(\varphi_{e}\right)\leq \delta +\frac{\delta }{2}. \end{array}\right.$$
(26)$$\mu_{y_{5}}\left(f\left(\varphi_{e}\right)\right)=\left\{\begin{array}{cc} \frac{4\;\cdot \;f\left(\varphi_{e}\right)-5\delta }{\delta }, & \delta +\frac{\delta }{4}<f\left(\varphi_{e}\right)\leq \delta +\frac{\delta }{2};\\ 1, & f\left(\varphi_{e}\right)\geq \delta +\frac{\delta }{2}. \end{array}\right.$$

Depending on the range in which the value of $f\left(\varphi_{e}\right)$, falls, one or another membership function $\mu_{y}$ is chosen with respect to the degree δ of decision-making. The degree δ belongs to the interval $\left[0;1\right]$ and is adjusted by the project analyst, and if necessary, it can be changed. This setting has the advantage that the model is easily adapted for a variety of grant projects and competitions, from student to multimillion H2020. Since the constructed membership functions (22)–(26) have intersections, for the evaluated projects $p_{e},\;e=\bar{1,n}$ we obtain either one or two levels of decision-making Y and, accordingly, the same number of reliabilities for them.

As a result of the calculation, we obtain the linguistic significance of the level of decision-making of the feasibility of financing project *Y* and its assessment of reliability. That is, the reliability of the fact that the evaluation of the project belongs to one or another level. Based on the initial data, investors make decisions on the appropriateness of financing projects to improve the sustainability and health of regions and cities, considering the goals of investors G  and the conclusions of experts L. If a situation arises where investors are not satisfied with any of the solutions, it is recommended to re-evaluation with additional data.

## 3. Results

The project evaluation criteria (proposed by experts) in the individual steps for quantitative evaluation as well as for project risk assessment must correspond to the problems to which the submitted projects are directed: the resilience and health of regions and cities within the European Green Deal and the European Industry 5.0 concept. Expert criteria will allow evaluating the content and the following benefits and impacts of the proposed projects (alone or in combination).

$K_{P}$—information models of criteria (groups of criteria) to assess the importance of the project idea to improve the sustainability and health of regions and cities.

A set of criteria is offered for assessing the importance of the project idea to improve the sustainability and health of regions, which is divided into five groups $C=\left(C_{1},C_{2},\ldots ,C_{5}\right)$. The evaluation criteria in each group *C* are presented in the form of a question to justify the importance of the project idea, where the expert, based on the read project application, scores from the interval [1; 20] for each criterion according to the proposed fuzzy rules.

Group $C_{1}$—relevance, innovation, uniqueness of the project:

$K_{11}$—How does the project relate to the priorities for improving the resilience and health of the regions, namely:Sociocultural (activities or technologies that contribute to the preservation of cultural heritage, to the increase of citizens education in the regions or in cities, down extremism, racism and religious tolerance, support for marginalized groups, gender equality, equal treatment of men and women at work, in community and in the separation of labor, and other similar projects);Political (project activities to support the development of civil society and active citizen participation in public affairs, activities supporting citizen co decision in regional and urban development plans, activities supporting citizen participation in the implementation of policies at regional and local level, activities supporting and improving regional and urban management, support for the resilience of society and citizens against misinformation in the public and digital space, support for corporate social responsibility and active citizenship programs to meet the objectives of the European Green Deal and the European Industry 5.0 concept, strengthening the influence of the third sector, volunteering and cities, and other similar projects);Economic (project impact on job creation in the region and cities, improving the business environment and business development in regions and cities, impact on local taxes, project investments in regional and urban development—global resources, regional resources, local resources, private funding sources, development and investment projects, technologies and procedures reducing energy intensity, operation, and maintenance of buildings and infrastructure in regions and cities within the public and private sector, reducing the financial costs of regional and city administration, support for citizens’ financial literacy programs, and other similar projects); Environmental (type, number, and extent of green technologies and applied procedures for green regions and cities, projects for the application of alternative energy sources, impact on water supply and water pollution in regions/cities on the climate, impact on soil decontamination, reduction of air pollution sources in regions and cities, projects with an impact on food safety and food self-sufficiency, projects to increase companies’ awareness of their impact on the environment, projects to improve the quality of the indoor environment of public and private buildings, and other similar projects);Security (aspects of the project strengthening the resilience of the region and cities for security periods also in crisis situations, in preparing citizens for crisis situations, in preparing and improving the quality of human, material, and technical resources for crisis situations in regions and cities, benefits of business and non-business entities, civic associations and charities in regions and cities to prepare citizens and society for crisis situations, and other similar projects); Health (climate change also increases the risk of future pandemics and endangers the quality of life of citizens, therefore we evaluate the benefits and impacts of proposed projects to improve lifestyle health, equality, and equity of available health services and facilities in regions and cities, improve infrastructure of health facilities in regions and cities, or modernization projects to improve their services to citizens, non-investment projects to improve services to citizens under, and other similar projects).

$K_{12}$—completeness of definition, originality, and validity of the main ideas, proposals, sequence of development, which can be meaningful, original, and justified based on world experience; meaningful, original, and justified based on national experience; reasonable, but mostly consider the experience of performers; declarative, but not justified. 

$K_{13}$—novelty of expected results, their difference from existing developments, completeness of disclosure and analysis of analogues and prototypes, namely in the application: well-defined expected results, revealed their differences from existing world analogues; novelty of results only at the national level; novelty of project results only at the local level. 

$K_{14}$—the validity of the importance of the project for the applicant organization and partners, given the main/strategic activities. It considers the previous experience of the applicant and the partner for the project.

Group $C_{2}$—goals, objectives, short-term results of the project:

$K_{21}$—the application demonstrates the logical principle of the causal link between the definition of the purpose, objectives, objectives, and results of the project. In the application, all components are interdependent and subordinate; indicates how the partner is involved in the implementation of goals and objectives.

$K_{22}$—the results of the project achievement can be tracked within the project or immediately after its completion, the intermediate and final goals of the achievement are related to the defined objectives of the project.

$K_{23}$—the quality of the work plan of the project, which is identical to the application and estimate without factual differences. The work plan reflects in detail and in a strict logical sequence all the main stages of project implementation, indicates the activities of the partner organization, contains all the necessary components such as implementation stages, results, responsible persons, organizational and economic forms of stage executors. 

Group $C_{3}$—is the target audience of the project:

$K_{31}$—target groups are clearly defined and correctly described through quantitative and qualitative indicators. Their needs have been identified in advance, or information on the existence of these needs that has been properly substantiated, possibly with reference to research. Target audiences of the project meet the goals, objectives, relevance of the project is confirmed by the needs and interests of target audiences.

$K_{32}$—the value of the project to the target audiences is clearly defined, it explains how the project meets the cultural needs and interests of the target audiences, identifies stakeholders, and describes how stakeholders will interact with the project or its results; the uniqueness and innovation of the project is confirmed by the needs and interests of the target audiences. It indicates how the applicant will work with stakeholders.

$K_{33}$—the communication plan of the project helps to draw attention to the results of the project on improving the resilience and health of regions, forming a sense of target audiences to the importance of sustainability and health, both at local and regional levels.

Group $C_{4}$—long-term project results:

$K_{41}$—What long-term results will be achieved through the project implementation? What will confirm the achievement of the project goal? The applicant aims to achieve the effect of long-term impact on its target audiences, environment, is the impact that can be fully realized in three to five years, this impact is adequately described through the indicators of project results.

$K_{42}$—the applicant provides measures for public presentation of project results in accordance with the project objectives and selected target audiences. 

$K_{43}$—further activities are planned to prolong the long-term impact of the project: free access to project results, free access to project information. 

$K_{44}$—the applicant organization plans to share its experience with other organizations, plans to establish partnerships with other organizations outside the project to further develop the idea.

Group $C_{5}$—quality of the project estimate:

$K_{51}$—the quality of cost estimates for compliance with the specified project objectives and investor requirements.

$K_{52}$—compliance of the budget with the stated objectives of the project. The project work plan, tasks and specific results are correlated with the estimate, as well as with the expected results. 

$K_{53}$—cost-effectiveness is that the ratio between costs and expected results is satisfactory and rational, this one that confirms the efficient and transparent use of funds for project implementation.

$K_{54}$—the validity of costs is that all budget items are written in the form of formal records: price per unit, number of units, number of months, kilometers, number of participants, etc., as a result—the formation of prices for a single item of expenditure is transparent, reasonable. For example, travel expenses, attraction of material and technical base, administrative and other expenses are motivated by the goals and objectives of the project, contribute to their implementation, and do not contradict the requirements and restrictions of investors.

The above set of criteria is open, and the model does not depend on the number of groups. Investors can always add their own indicators when considering specific projects to improve the resilience and health of regions.

$K_{R}$—information models of criteria for assessing risk-oriented factors of influence that will potentially lead to the failure of the project.

The issue of assessing risk-oriented factors influencing project implementation is very complex. Depending on the project, the region, and the stages of project implementation, different risk indicators need to be adjusted. There are many classification approaches to risk assessment for both classic and innovative and start-up projects. Here are some criteria by which the expert can assess the risks that may arise in the implementation of projects to improve the sustainability and health of regions and cities:

$K_{R1}$—risks of insufficient consideration of environmental factors of the project, prospects for its completion and development;

$K_{R2}$—risks of insufficient consideration of factors of behavior of competitors;

$K_{R3}$—risks of unforeseen expenses and reduced income;

$K_{R4}$—risks of project failure due to unforeseen budget changes and changes in funding levels;

$K_{R5}$—risks associated with innovations declared in the project implementation plans to increase the sustainability and health of regions and cities;

$K_{R6}$—risks are related to insufficient awareness of staffing with the scope of the project;

$K_{R7}$—environmental risks associated with not achieving the goals within the Green Deal concept; 

$K_{R8}$—marketing risks at different stages of project implementation;

$K_{R9}$—marketing risks at the stage of presentation of results (sales of work results) of the project;

$K_{R10}$—risks of similar competitive projects in the region of project implementation;

$K_{R11}$—risks associated with securing property rights to innovations, patents.

The above set of criteria is completely open and inexhaustible. For example, if a thematic project selection competition is held, then project analysts must adapt this set of criteria to the theme of this competition.

According to $K_{T}$—information models of criteria (groups of criteria) for assessing human factors and the team of project implementers, their experience, and knowledge in the field of sustainability and health of regions and cities, we propose to use information models already developed by the authors. For example, if the project is submitted by a team of developers, then you can use the information model of evaluation and rating of the team of developers of start-up projects [38]. If the project is represented by an organization, then one of the approaches can be used [37].

The result of the study, the algorithm of the model, was tested on the example of the evaluation of five submitted projects $P=(p_{1};p_{2};\ldots ;p_{5})$ to improve the sustainability and health of regions and cities, which will be implemented by the Regional Development Agency of the Transcarpathian Region (Ukraine). For the creation of the model algorithm, the knowledge and good practice of the authors were used as experts and evaluators of the 896 projects from the stock of projects of the Partnership Council of the Trnava self-governing region in Slovakia, submitted under the plan of economic development and social development for implementation and financing in 2021–2027.

The calculations will be performed based on the developed complex hybrid model of project evaluation to improve the sustainability and health of regions and cities. To do this, we will evaluate separately on fuzzy models $M_{P}$, $M_{R},$ $M_{T}$, and $M_{A}$—the model of aggregation of output data for deriving the level of decision-making expediency of project financing. The evaluation was conducted by the authors of the article, who are experts in various commissions and competitions for the evaluation of grants, scientific, technical, and start-up projects.

For example, consider in more detail the evaluation of the project $p_{1}$, which is a sociocultural project that was successfully implemented in 2021 in Ukraine—“Showcase of Zakarpattia” [39], on models $M_{P}$ and $M_{R}$.

$M_{P}$—fuzzy project evaluation model to improve the sustainability and health of regions and cities.

At the first stage of the project $p_{1}$ we get input data for each evaluation criterion according to $K_{P}$—information models of criteria (groups of criteria) to assess the importance of the project idea to improve the sustainability and health of regions and cities. Suppose that the expert sets his statements on some interval of numbers $x\in \left[1;20\right]$ using one of the fuzzy statements *RL =* {*A*; *B*; *C*; *D*}. For each criterion and group of criteria, the project analyst determined the weights. Input data and weights are given in Table 1.

At the next stage, we will move from the intervals of fuzzy project evaluations according to the criteria of one group, to one-point evaluation within the group. To do this, first, based on the input data, using the membership functions of criteria (2)–(5) we construct graphs of expert opinions. Next, the weights were divided within the corresponding group, according to Formula (6). The result is presented in the form of a vector: $\alpha_{1}$= (0.28; 0.26; 0.26; 0.2), $\alpha_{2}$= (0.35; 0.35; 0.3), $\alpha_{3}$= (0.39; 0.35; 0.26), $\alpha_{4}$= (0.28; 0.22; 0.25; 0.25), $\alpha_{5}$= (0.28; 0.25; 0.22; 0.25). To fuzzification the data, we find the weighted sum by Formula (7). The results of the calculation, separately for the groups of criteria, will be illustrated on the graphs of functions that are built using our designed software (Figure 3).

For the aggregate assessment of the expert opinion $x_{i}\in \left[1;20\right]$ of the corresponding group of criteria  i, we take the maximum value of the membership function of the weighted sum: $\underset{x}{\max}\bar{\varepsilon_{1}}\left(x\right)$ = 0.93 then $x_{1}$ = 15; $\underset{x}{\max}\bar{\varepsilon_{2}}\left(x\right)$ = 0.91 then $x_{2}$ = 13; $\underset{x}{\max}\bar{\varepsilon_{3}}\left(x\right)$ = 0.95 then $x_{3}$ = 16; $\underset{x}{\max}\bar{\varepsilon_{4}}\left(x\right)\;$= 0.9 then $x_{4}$ = 13; $\;\underset{x}{\max}\bar{\varepsilon_{5}}\left(x\right)\;$= 0.97 then $x_{5}$ = 14.

Thus, an aggregate conclusion of the expert’s opinions on the criteria for each group was obtained.

In the third stage, the aggregated conclusions of the experts on the groups of criteria will be combined into a general assessment. To do this, calculate the normalized weights for each group of criteria, according to the Formula (8): $\alpha_{1}=0.22;$ $\alpha_{2}=0.16;$ $\alpha_{3}=0.2;$ $\alpha_{4}=0.22;$ $\alpha_{5}=0.2.$ For defuzzification data, we construct an aggregate estimate using a weighted average convolution according to the Formula (9): $m_{p}=\frac{1}{20}\left(15\cdot 0.22+13\cdot 0.16+16\cdot 0.2+13\cdot 0.22+14\cdot 0.2\right)$ = 0.71.

$M_{R}$—fuzzy model for assessing the risks of project implementation to improve the sustainability and health of regions and cities.

At the first stage of the project $p_{1}$ we receive input from the expert on each evaluation criterion, according to the proposed $K_{R}$—information models of criteria for assessing risk-oriented factors of influence that will potentially lead to the failure of the project. Input data and weights according to the criteria determined by the project analyst are given in Table 2.

In the first stage, we will carry out fuzzification of input hybrid data. To do this, the values of $O_{u}$, $\varepsilon_{u}$ and $\mu \left(x_{u}\right)$ were calculated by the Formula (11). In the second stage, we calculate the normalized weights by Formula (12). The results of the calculation of the first and second stages are shown in Table 3.

In the third stage, we derive an aggregate risk assessment according to the Formula (13): $m_{r}$ = 0.63.

$M_{T}$—fuzzy model for assessing the competencies of the project implementation team.

The team of project implementers $p_{1}$, their experience and knowledge in the field of sustainability and health of regions and cities are proposed to evaluate according to the Information model of evaluation and output rating of start-up projects development teams where all data on the calculation procedure are given in [37]. After the expert evaluation, we received that the teams implementing the $p_{1}$ project received the following number of points: $m_{t}$ = 0.74. 

Thus, normalized estimates of the project $x_{1}$ according to fuzzy estimation models were obtained $m_{p}\left(p_{1}\right)=0.71,\;m_{r}\left(p_{1}\right)=0.63,\;m_{t}\left(p_{1}\right)=0.74$.

Similarly, the other projects are calculated, for which we obtain the following estimates: $m_{p}\left(p_{2}\right)=0.62,\;m_{r}\left(p_{2}\right)=0.87,\;m_{t}\left(p_{2}\right)=0.42;$ $m_{p}\left(p_{3}\right)=0.77,\;m_{r}\left(p_{3}\right)=0.82,\;m_{t}\left(p_{3}\right)=0.52;\;m_{p}\left(p_{4}\right)=0.51,\;m_{r}\left(p_{4}\right)=0.58,\;m_{t}\left(p_{4}\right)=0.74;$ $m_{p}\left(p_{5}\right)=0.81,\;m_{r}\left(p_{5}\right)=0.56,\;m_{t}\left(p_{5}\right)=0.36$.

Let the investor have his own goals regarding the need and possibility of financing projects to improve the sustainability and health of regions and cities.

If the investor needs the importance of the region where the project will be implemented $A_{1}=\left\{2\;\mathrm{category}\;\mathrm{of}\;\mathrm{the}\;\mathrm{region}\right\}$, the acceptable risk $A_{2}=\left\{\mathrm{low}\;\mathrm{risk}\right\}$ and the competence of the project implementation subjects $A_{3}=\left\{\mathrm{interested}\;\mathrm{in}\;\mathrm{average}\;\mathrm{competencies}\right\}$ then $U=\left(0.8,\;0.6,\;0.7\right)$.

Next, for all projects the values of $Z_{e}=\left(z_{pe},z_{re},z_{te}\right),\;e=\bar{1,5}$ were found, which characterize the relative estimates of the proximity of the evaluated projects to the vector of investors’ goals, for (17)–(19). For example, for a project $p_{1}$: $z_{p1}=1-\frac{\left|0.8-0.71\right|}{max\left\{0.8-0.51;0.81-0.8\right\}}=0.69,$ $z_{r1}=1-\frac{\left|0.6-0.63\right|}{max\left\{0.6-0.56;0.87-0.6\right\}}=0.889,$ $z_{t1}=1-\frac{\left|0.7-0.74\right|}{max\left\{0.7-0.36;0.74-0.7\right\}}=0.882$.

Next, to aggregate the values of $Z_{e}$ we use the three-dimensional cone-like membership function by the Formula (20). 

After that, the experts evaluating the projects express their conclusions on the possibility of achieving the goal of improving the sustainability and health of regions and cities, by implementing this project with the support of investors and considering their goals. 

Next, the dependencies of the initial assessment $\varphi_{e}$ and of the expert opinion L on the possibility of achieving the goal of improving the sustainability and health of regions and cities, by implementing this project with the support of investor goals according to the Formula (21) were interpreted. 

All the results of calculations and conclusions of experts *L* are presented in Table 4.

Thus, aggregate normalized estimates $f\left(\varphi_{e}\right),\;e=\bar{1,5}$. for project evaluation models, investor goals, and expert opinions evaluating projects were obtained. 

Next, consider $M_{A}$—the model of aggregation of output data for deriving the level of decision-making expediency of project financing. To obtain the output estimate $f=\mu_{Y}\left(f\left(\varphi_{e}\right)\right)\;$ and the decision−making level Y, which contains the content of the feasibility of project financing, considering the goals of investors G  and the conclusions of experts L, we use triangular membership functions (22)–(26). Let the project analyst determine the degree δ=0.6, then we get the following levels for the five evaluated projects:
$p_{1}$: decision-making level $y_{5}$ with output estimate $\mu_{y_{5}}\left(0.969\right)$ = 1;$p_{2}$: decision-making level $y_{1}$ with output estimate $\mu_{y_{1}}\left(0.443\right)=0.05$ or decision-making level $y_{2}$ with output estimate $\mu_{y_{2}}\left(0.443\right)=0.95$;$p_{3}$: decision-making level $y_{3}$ with output estimate $\mu_{y_{3}}\left(0.611\right)$ = 0.93 or decision-making level $y_{4}$ with output estimate $\mu_{y_{4}}\left(0.611\right)=0.07$;$p_{4}$: decision-making level $y_{5}$ with output estimate $\mu_{y_{5}}\left(0.903\right)$ = 1;$p_{5}$: decision-making level $y_{2}$ with output estimate $\mu_{y_{2}}\left(0.598\right)$ = 0.01 or decision-making level $y_{3}$ with output estimate $\mu_{y_{3}}\left(0.598\right)$ = 0.99.

Based on the output data, investors decide on the feasibility of financing projects to improve the sustainability and health of regions and cities, within the concepts of the European Green Deal and Industry 5.0, considering the goals of investors *G* and expert opinions L. 

As shown, two projects $p_{1}$ and $p_{4}$ received $y_{5}$—a very high level of feasibility of financing the project with a reliability of 1. Project $p_{1}$ has already been successfully funded and implemented, in line with the objectives of the European Green Deal. The knowledge gained because of the $p_{1}$ project became a model of succession and was also the basis for the presentation of subsequent innovative projects for other regions.

## 4. Discussion

The paper presents a comprehensive hybrid model for evaluating projects to improve the sustainability and health of regions and cities, within the concepts of the European Green Deal and Industry 5.0. To this end, the following have been developed: information models of input data for the evaluation of projects to improve the sustainability and health of regions and cities; fuzzy project evaluation model; fuzzy model for assessing the risks of project implementation; model of aggregation of output data for deriving the level of decision-making expediency of project financing. 

The complex hybrid model can adequately determine the level of feasibility of project financing, considering the goals of investors and expert opinions on the possibility of achieving goals to improve the sustainability and health of regions and cities through the project. The study is based on the apparatus of fuzzy sets based on estimation intervals, showing the corridor of values of forecast parameters. This allows for increasing the degree of validity of decisions, as it considers all possible development scenarios, depicting a continuous spectrum. For the processing of expert information and fuzzy input data, intelligent analysis of knowledge is also used based on the membership functions to the evaluation criteria, one, and many variables, considering any type of input data. Intellectual analysis of knowledge allows revealing the subjectivity of experts and to obtain a quantitative assessment of an informal applied problem. The models developed in the work reveal the vagueness of the incoming expert opinions and increase the degree of validity of further decisions by investors on the choice of project for its financing. The value of the model is that it allows obtaining a comprehensive quantitative assessment of the project based on descriptive input (text) data obtained from the project application. For the expert, the evaluation procedure remains classic and well-known, he examines the project application, then on several issues expresses his views on the importance of the idea and quality of the project. After that, the data are processed by appropriate fuzzy and hybrid models, revealing the subjectivity of experts, and adjusting the parameters of the models and the target needs of investors to prevent the subjective influence of participants in the evaluation process on the result. At the end of the model, is the output quantitative assessment and linguistic significance of the level of decision-making expediency of project financing with the assessment of reliability.

The advantages of a complex hybrid model for evaluating projects to increase the sustainability and health of regions and cities, within the concept of the European Green Deal and Industry 5.0, stem from the advantages of the developed models. The hybrid model is based on various information models of input data, adapted to evaluate projects to improve the sustainability and health of regions and cities, and fuzzy models to assess various aspects of the presented projects, from the idea, and implementation risks to the contractors. The set of criteria is open, the model does not depend on their number, and project analysts can always adapt the set of criteria to highly specialized project topics. The model considers the investor’s goals regarding the need and possibility of financing projects, namely: the purpose of the importance of the region where the project will be implemented, the purpose of acceptable risks, and the purpose of competence of project participants. The developed fuzzy knowledge base for evaluating projects to improve the resilience and health of regions and cities can be easily adapted to different goals of investors. The quality of the final decision is improved by involving an expert opinion on the possibility of achieving the goal of improving the sustainability and health of regions and cities, by implementing this project with the support of investors, and by considering their goals. The hybrid model determines the level of expediency of project financing, and there is a possibility, depending on the need, to change the degree of decisions. Models reveal the vagueness of input estimates, increase the degree of validity of future decisions, and focus on the impartial evaluation of projects, which in turn increases the security of their financing.

The key result of the presented paper is a comprehensive approach to the evaluation of projects to improve the sustainability and health of regions and cities, within the European Green Deal and Industry 5.0 concepts, for practical use by evaluation commissions and decision-makers at national and local level, or private sector levels. The final algorithm of the hybrid model and its verification of applicability for the evaluation of projects in the selected area of interest of the grant provider confirmed the validity of hypothesis H1 and also confirmed the validity of hypothesis H2. In the context of the European Green Deal goal to reduce the net greenhouse gas emissions to zero by 2050, the novelty of Simionescu et al.’s paper [16] is related to the effective proposal of measures to improve quality of governance to achieve this goal in the Central and Eastern European countries (represented by Bulgaria, Czech Republic, Estonia, Hungary, Latvia, Lithuania, Poland, Romania, Slovak Republic, and Slovenia). The resulting hybrid model is a practical tool to support the implementation of state policy in the framework of objective, the transparent and anti-corruption policy at the regional level in the expert evaluation of projects to strengthen the resilience and health of regions and cities. The study by Fidlerová et al. [22] provides evidence of the perception of the business world from six countries (Finland, Slovakia, Italy, Austria, Spain, and Turkey) of their intention to identify business opportunities through sustainable development goals in different countries and sectors through strategy and practice. The created hybrid model enables the evaluation of such a potential of the innovative ecosystem of companies in business opportunities within the framework of strengthening the resilience and health of regions and cities. Multi-criteria analysis of Chovancova et al. [24] has thrown a spotlight on the Achilles heel of the EU country’s disregard which might cause serious problems in the future for ensuring universal access to modern energy services, improving energy efficiency, and increasing the share of renewable energy. The created hybrid model based on fuzzy expert evaluation of projects allows obtaining high ratings in innovative proposals for strengthening regional resilience in the field of energy security and diversity of resources and solutions, with minimal negative impacts on public health and the environment. A survey by a team of authors Maddikunta et al. [17] on supporting technologies and potential applications in Industry 5.0 provides a discussion of smart healthcare, cloud manufacturing, supply chain management and manufacturing, some supporting technologies for Industry 5.0, such as edge computing, digital twins, collaborative robots, the Internet of all things, blockchain and 6G networks and beyond. The created hybrid model based on fuzzy expert evaluation of projects allows obtaining high ratings in innovative proposals for strengthening the health of regions and cities, developing health infrastructure and improving the quality of health services provided using artificial intelligence for prevention, individual health counseling and evidence-based medicine support. In a study by Xu et al. [18] Industry 4.0 is considered to be a technology-based industry, while Industry 5.0 is a value-based industry. The coexistence of two industrial revolutions raises questions and therefore requires discussion and clarification. The resulting hybrid model is a practical tool to support decision-making processes in the expert evaluation of projects to strengthen the resilience and health of regions and cities, appreciating solutions to human and technological aspects of transfer to the Industry 5.0 concept and the potential to manage them. The study by Sindhwani et al. [19] notes that the Industry 5.0 revolution is a challenge to put the ideas of sustainability into practice, to integrate human values with technology and is considered a step forward in achieving the goals of sustainable development. This study therefore proposes a framework for analyzing the factors that enable Industry 5.0 to achieve sustainability integration of human values with technology. The study inspires the creators of the hybrid model with the main focus criteria in Industry 5.0, which support resilience and societal value creation, and a new framework that combines the four steps needed to solve any Multi-Criteria Decision-Making problem: selection, weight, ranking, and verification. The created hybrid model based on fuzzy expert evaluation of projects allows obtaining high ratings in proposals to strengthen the resilience and health of regions and cities, which is an innovative way to address the integration of human values with advanced technologies.

The disadvantages (limitations) of this model include the use of different types of membership functions, namely: for fuzzy *RL* statements, to combine conclusions about the level of probability of risk situation and the number of confidences of the expert’s opinions on providing their opinion, to aggregate $Z_{e}$, o interpret the dependence estimates $\varphi_{e}$ and expert opinion L. The choice of the type of membership functions and the use of different types of convolutions can lead to ambiguity in the results. 

The rationality of the obtained assessment of the level of expediency of project financing proves the advantages of the developed model. The reliability of the obtained results is ensured by the reasonable use of the apparatus of fuzzy sets, intellectual analysis of knowledge, and systematic approach, which is also confirmed by the results of the research.

## 5. Conclusions

To develop a complex hybrid model for evaluating projects to strengthen the resilience and health of regions and cities, the authors used their experience with a team of researchers and members, respectively: chairman of the working groups for transport and health infrastructure within the activities and plans for economic development and social development of the selected regions. Findings from the stack of project intentions, 896 projects in the field of transport, influenced the creation of an algorithm for solving the problem. Seven invited experts with more than 15 years of experience in project management helped determine the evaluation criteria. The algorithm of the expert model was verified using five real projects. The gained experience enables the transfer of knowledge and the methodology of solving the problem has the potential for repetition by other researchers, resp. project evaluators in practice. The new contributions and new understanding in the field based on the research of the topic are as follows:For the first time developed information models of input data for evaluation of projects to improve the sustainability and health of regions and cities. The set of criteria is open, and the model does not depend on their number. Project analysts can always adapt many criteria to highly specialized project topics;The model uses an adequate apparatus of fuzzy sets and allows obtaining a quantitative assessment of the project based on input descriptive (textual) data obtained from the project application, which increases the degree of validity of future decisions;The model reveals the vagueness of input estimates and can derive a quantitative normalized initial value of the risks of the project, which increases the degree of validity of further management decisions;The model determines the level of feasibility of project financing, considering the target needs of investors and expert opinions on the possibility of achieving goals to improve the sustainability and health of regions and cities through the implementation of this project. The hybrid complex model focuses on the impartial evaluation of projects and increases the security of their financing;The model is easily adapted for different size grant projects and competitions;The results demonstrate the applied value of the methodology for assessing the level of decision-making, the feasibility of project financing in conditions of uncertainty, and the importance of making sound management decisions based on expert opinions and unclear conditions.

The created hybrid model has the potential for implementation in the following key areas. The model can be used to support the decision-making processes of evaluation commissions at the level of public administration and self-government and entities authorized to implement regional policy and local self-government with the support of specialized, targeted projects, resp. programs. Second, the model can be used in the environment of small and medium enterprises, respectively, in the non-commercial sphere. Third, the results of scientific research can also be used to educate and support young scientists and professionals with social responsibility, which is usually reflected in the creation of social responsibility of companies, institutions or public associations in an open civil society.

The following research will focus on the development of information technologies for the evaluation of projects to improve the sustainability and health of regions and cities, within the concepts of the European Green Deal and Industry 5.0. The integration of human values with advanced technologies remains a major challenge. Information technologies will be based on the created hybrid model and software for project evaluation. Information technology, the hybrid model and software will together support decision-making on the security of grant funding. The developed model and its software support will be a useful tool for project analysts in preventing inefficient project financing and supporting the European Green Deal and Industry 5.0 concepts. Addressing this issue is also of great importance for countries applying for membership of the European Union, such as Ukraine. A state devastated by the war will have to rebuild society and the economy, while transparency of processes and independent expert evaluation of projects remain a constant challenge.

## Figures and Tables

**Figure 1 ijerph-19-08217-f001:**
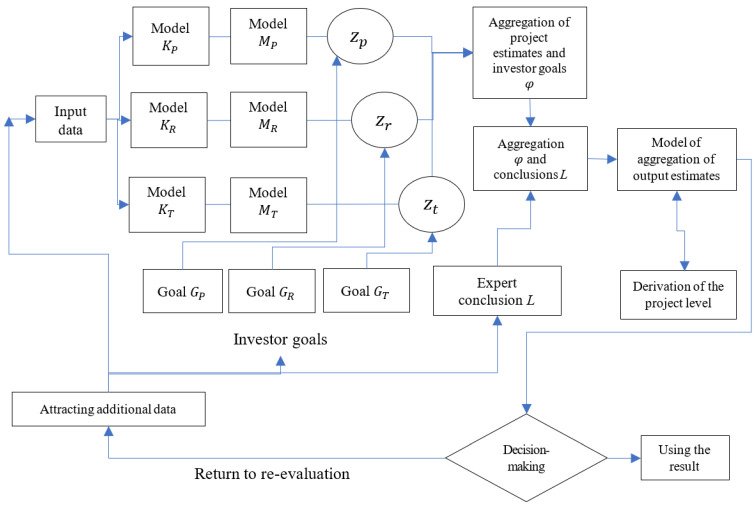
Block diagram of the complex hybrid model.

**Figure 2 ijerph-19-08217-f002:**
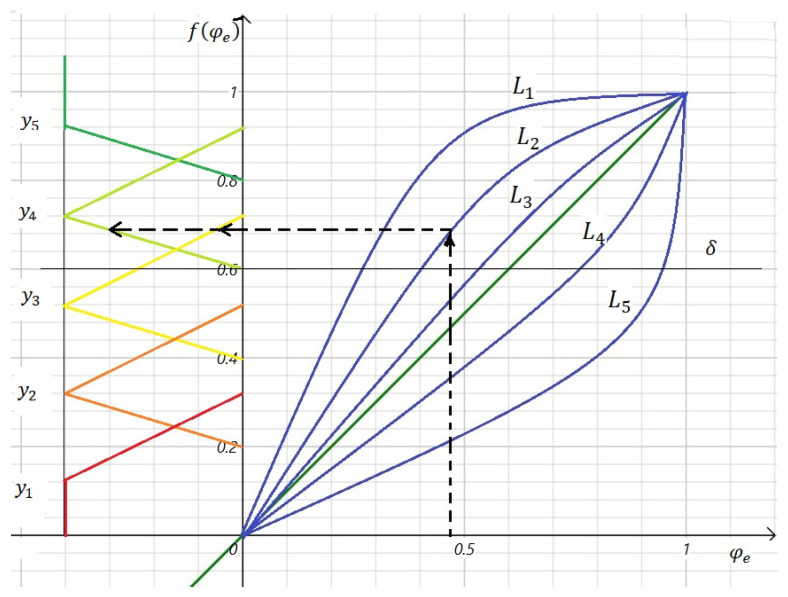
Graphic interpretation of the model $M_{A}$ (*y*—levels of decision-making; *L*—expert opinion; $\varphi_{e}$—initial estimates; $f\left(\varphi_{e}\right)$—aggregate normalized estimate; δ—degree of decision making).

**Figure 3 ijerph-19-08217-f003:**
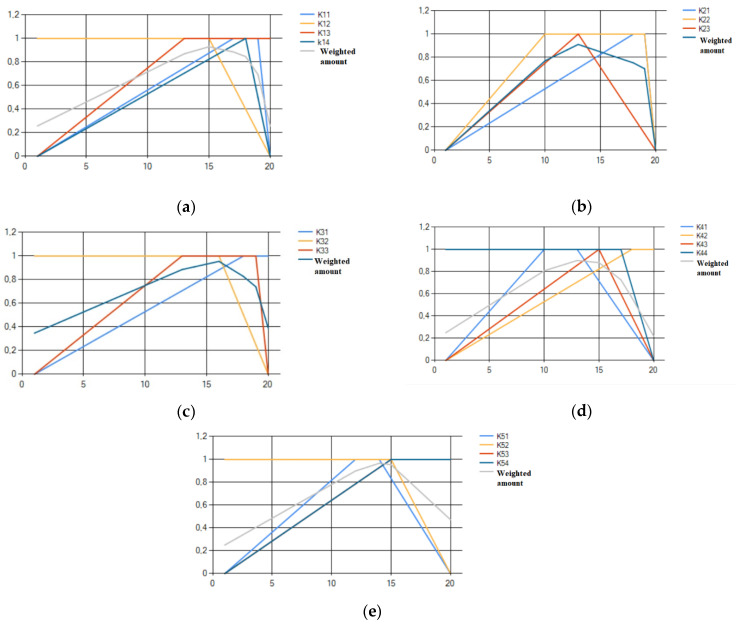
The result of the calculation of the project p_1 according to the model $M_{P}$: (**a**) the weighted amount for a group of criteria $C_{1}$; (**b**) the weighted amount for a group of criteria $C_{2};\;$ (**c**) the weighted amount for a group of criteria $C_{3};$ (**d**) the weighted amount for a group of criteria $C_{4};$ (**e**) the weighted amount for a group of criteria $C_{5}$.

**Table 1 ijerph-19-08217-t001:** Input data on expert evaluation according to $K_{P}$.

Group Criteria	Name Criteria	Fuzzy Statements	Evaluation Limits *x*	Weight Groups	Weight Criteria
$C_{1}$	$K_{11}$	A	from 17 to 19	10	10
	$K_{12}$	C	not more than 15		9
	$K_{13}$	D	not less than 13		9
	$K_{14}$	B	close to 18		7
$C_{2}$	$K_{21}$	A	from 18 to 19	8	7
	$K_{22}$	A	from 10 to 19		8
	$K_{23}$	B	close to 13		8
$C_{3}$	$K_{31}$	D	not less than 18	9	9
	$K_{32}$	C	not more than 16		8
	$K_{33}$	A	from 13 to 19		6
$C_{4}$	$K_{41}$	A	from 10 to 13	10	9
	$K_{42}$	D	not less than 18		7
	$K_{43}$	B	close to 15		8
	$K_{44}$	C	not more than 17		8
$C_{5}$	$K_{51}$	A	from 12 to 14	9	10
	$K_{52}$	C	not more than 15		9
	$K_{53}$	D	not less than 15		8
	$K_{54}$	A	from 15 to 20		9

**Table 2 ijerph-19-08217-t002:** Input data on expert evaluation according to $K_{R}$.

Name Criteria	Conclusions Experts	Confidence of the Expert’s Reasoning	Weight Criteria
$K_{R1}$	$t_{1}$	0.8	10
$K_{R2}$	$t_{2}$	0.9	9
$K_{R3}$	$t_{1}$	0.7	9
$K_{R4}$	$t_{1}$	0.9	7
$K_{R5}$	$t_{3}$	0.7	10
$K_{R6}$	$t_{3}$	0.7	9
$K_{R7}$	$t_{1}$	0.8	10
$K_{R8}$	$t_{3}$	0.9	9
$K_{R9}$	$t_{2}$	0.9	8
$K_{R10}$	$t_{2}$	0.8	8
$K_{R11}$	$t_{1}$	0.8	9

**Table 3 ijerph-19-08217-t003:** Calculation results for the model $M_{R}$.

Name Criteria	$\mathrm{Value}\;O_{u}$	$\mathrm{Value}\;\varepsilon_{u}\;$	Membership $\mathrm{Function}\;\mu \left(K_{Ru}\right)$	Normalized Weight $\mathrm{Criteria}\;\;\bar{v_{u}}$
$K_{R1}$	84	0.256	0.744	0.102
$K_{R2}$	64	0.374	0.626	0.092
$K_{R3}$	86	0.331	0.669	0.092
$K_{R4}$	82	0.206	0.794	0.071
$K_{R5}$	58	0.516	0.484	0.102
$K_{R6}$	58	0.516	0.484	0.092
$K_{R7}$	84	0.256	0.744	0.102
$K_{R8}$	46	0.549	0.451	0.092
$K_{R9}$	64	0.374	0.626	0.082
$K_{R10}$	68	0.377	0.623	0.082
$K_{R11}$	84	0.256	0.744	0.092

**Table 4 ijerph-19-08217-t004:** Calculation results and expert opinions *L*.

	$p_{1}$	$p_{2}$	$p_{3}\;$	$p_{4}$	$p_{5}$
Evaluation $z_{p}$ relative to the goal $G_{P}$	0.690	0.379	0.897	0	0.966
Evaluation $z_{r}$ relative to the goal $G_{R}$	0.889	0	0.185	0.926	0.852
Evaluation $z_{t}$ relative to the goal $G_{T}\;$	0.882	0.176	0.471	0.882	0
Output estimates $\varphi_{e}$	0.883	0.521	0.674	0.663	0.663
Expert opinions *L*	$L_{1}$	$L_{4}$	$L_{4}$	$L_{1}$	$L_{4}$
Aggregate normalized estimates $f\left(\varphi_{e}\right)$	0.969	0.443	0.611	0.903	0.598

## Data Availability

The data presented in this study are available on request from the corresponding author.

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
