# Peer review of "A Complex Hybrid Model for Evaluating Projects to Improve the Sustainability and Health of Regions and Cities"

_ijerph, 2022, doi:10.3390/ijerph19138217_

Round 1

Reviewer 1 Report

Dear authors,

The article addresses a relevant topic, both from a scientific point of view and in terms of practical implications. However, for publication in a journal such as the International Journal of Environmental Research and Public Health, I recommend considering the following improvements:

- the submitted version has an excessively long title; a more concise alternative would be more suitable;

- throughout the text there are some disconnected words and punctuation; I recommend a linguistic review;

- in some passages of the text the authors present a subjective position; for instance, statements such as those presented in the first paragraph of the introduction require citation of relevant references so that they can be properly corroborated;

- It is necessary to consider that scientific work must have a formal and impersonal character; should prefer using third-person perspective instead of first-person;

 - in several passages there are repetitions of ideas; a revision is needed to make the text more concise, direct and fluid;

- abbreviations that appear for the first time in the text must be spelled out in full; for instance, SME’s;

- the topic materials and methods is very extensive, exceeding more than 10 pages; I recommend rethinking the way of presentation, making it more direct and clear to the reader;

- separate what was a working method from what is a methodological result; if the model developed is a result of the work, then it should be presented as such and not in the topic of materials and method;

- the presentation of the results of the study on the example of the evaluation of five projects is not clearly described, limiting the understanding of how the developed model was applied;

- no literature was addressed in the discussion of the results; the lack of rigor regarding the discussion section is a serious flaw for an international journal like IJERPH;

- I recommend that the Conclusions section should be rewritten to reflect new contributions and new understandings in the field based on your research - not a repeat of the Results.

Yours sincerely, reviewer.

Author Response

Dear Reviewer,

We appreciate your positive evaluation and we consider it to be a significant motivation for our further work. We have carefully considered your comments and incorporated them into the revised manuscript hoping that the changes are in line with your ideas. The responses to your comments and recommendations can be found in the following text and all changes in the revised manuscript are highlighted.

All rounds of the reviewing process and your reviews have been inspiring, and we see that this has greatly improved the quality of our article. We would like to thank you for the constructive feedback and for the opportunity to revise our manuscript.

Kind regards,

Beata Gavurova

The reviewer 1: Comments and suggestions to Authors:

  1. the submitted version has an excessively long title; a more concise alternative would be more suitable;

Article title has been abbreviated. Lines 3-4.

  1. throughout the text there are some disconnected words and punctuation; I recommend a linguistic review;

The authors ordered English Editing Services for language correction of the paper.

  1. in some passages of the text the authors present a subjective position; for instance, statements such as those presented in the first paragraph of the introduction require citation of relevant references so that they can be properly corroborated;

The statement in the first paragraph was supplemented by a citation of the relevant source: 36. The European Regional Development Fund. Lines: 35-36.

  1. it is necessary to consider that scientific work must have a formal and impersonal character; should prefer using third-person perspective instead of first-person;

The authors ordered English Editing Services for language correction of the paper, plus they made corrections in the lines: 222-223, 236-237, 248-249, 272-273, 304, 308, 348-349, 387-388, 405, 473, 479, 506-509, 514, 521-522, 553, 559, 577, 609, 621-622, 636, 736, 805-813, 837, 852, 869, 877-878, 897, 910, 913-914, 933-935, 967.

  1. in several passages there are repetitions of ideas; a revision is needed to make the text more concise, direct and fluid;

Repetition of ideas was removed in the lines: 205-211, 261-266, 631-633, 798-804, 1116-1172.

  1. abbreviations that appear for the first time in the text must be spelled out in full; for instance, SME’s;

The full text of the abbreviation SMEs (Small and Medium-Sized Enterprises) was added at line: 169.

  1. the topic materials and methods is very extensive, exceeding more than 10 pages; I recommend rethinking the way of presentation, making it more direct and clear to the reader;

Part 2 Materials and Methods was reorganized. The original text on the complex hybrid model has been moved to Part 1 Introduction, as subchapter 1.2. The revised Section 2 starts on line 324. The original text of point 2.3 on the group of criteria has been relocated to Section 3 Results. Lines: 572-743.

  1. separate what was a working method from what is a methodological result; if the model developed is a result of the work, then it should be presented as such and not in the topic of materials and method;

The comment was implemented as part of the reorganization of Section 2 Materials and Methods. The working method was separated from the methodological result. Section 2 is from line 324.

  1. the presentation of the results of the study on the example of the evaluation of five projects is not clearly described, limiting the understanding of how the developed model was applied;

For the creation of the hybrid model algorithm, the knowledge and good practice of the authors were used as experts and evaluators of the 896 projects from the stock of projects of the Partnership Council of the Trnava self-governing region in Slovakia, submitted under the plan of economic development and social development for implementation and financing in 2021-2027.

The result of the study, the algorithm of the hybrid model, was tested on the example of the evaluation of five submitted projects P = (p_1; p_2;…; p_5) to improve the sustainability and health of regions and cities, which will be implemented by the Regional Development Agency of the Transcarpathian Region (Ukraine).

Lines: 805-813.

  1. no literature was addressed in the discussion of the results; the lack of rigor regarding the discussion section is a serious flaw for an international journal like IJERPH;

In the Discussion section, a discussion on the results in selected literature of 6 scientific papers was added, in relation to the results of the presented expert hybrid model. Lines: 991-1045.

  1. I recommend that the Conclusions section should be rewritten to reflect new contributions and new understandings in the field based on your research - not a repeat of the Results.

The Conclusions section was rewritten for new contributions and understanding in the field based on the research, the primary areas of implementation of the study results and subsequent scientific work. Lines: 1068-1112.

Reviewer 2 Report

The research presented in the manuscript is very current and interesting, but ...

1. I definitely review a paper without hypotheses!?! Sorry, has it ceased to be fashionable to set up research hypotheses, or what? I would start from there. Then research is done, then, according to hypotheses, the achieved results are proven and a discussion is conducted with previously achieved results. Or this is an example of the absence of methodological education. I consider this to be a great shortcoming of this paper (research, if done as such) and I think it must be resoleved.

2. Unfortunately, there are too few literary references to the material that deals with the concept of Industry 5.0, and I think that MUST be worked on. Only one (?!?) reference analyzes this term directly, and a couple touches on this concept. At least as far as I know, this concept is not so poorly covered in the literature because I myself have been researching this topic. All clear, the work is set wider, but, come on, just one!

Author Response

Dear Reviewer,

We appreciate your positive evaluation and we consider it to be a significant motivation for our further work. We have carefully considered your comments and incorporated them into the revised manuscript hoping that the changes are in line with your ideas. The responses to your comments and recommendations can be found in the following text and all changes in the revised manuscript are highlighted.

All rounds of the reviewing process and your reviews have been inspiring, and we see that this has greatly improved the quality of our article. We would like to thank you for the constructive feedback and for the opportunity to revise our manuscript.

Kind regards,

Beata Gavurova

The reviewer 2: Comments and suggestions to Authors:

  1. I definitely review a paper without hypotheses!?! Sorry, has it ceased to be fashionable to set up research hypotheses, or what? I would start from there. Then research is done, then, according to hypotheses, the achieved results are proven and a discussion is conducted with previously achieved results. Or this is an example of the absence of methodological education. I consider this to be a great shortcoming of this paper (research, if done as such) and I think it must be resoleved.

Hypothesis H1 and hypothesis H2 were added in Section 1. Introduction. Lines: 58-71.

  1. Unfortunately, there are too few literary references to the material that deals with the concept of Industry 5.0, and I think that MUST be worked on. Only one (?!?) reference analyzes this term directly, and a couple touches on this concept. At least as far as I know, this concept is not so poorly covered in the literature because I myself have been researching this topic. All clear, the work is set wider, but, come on, just one!

In part 1. Introduction, to the mentioned literature on Industry 4.0, resp. on the transition to Industry 5.0, another 4 scientific papers were added, which examined the issue of Industry 5.0. Lines: 150-159.

Round 2

Reviewer 2 Report

Thank you for accepting my suggestions.